# Recurrence of Oral Leukoplakia after CO_2_ Laser Resection: A Prospective Longitudinal Study

**DOI:** 10.3390/cancers14215455

**Published:** 2022-11-06

**Authors:** Adela Rodriguez-Lujan, Pia López-Jornet, Eduardo Pons-Fuster López

**Affiliations:** 1Colaborate of Medicine and Odontology, Hospital Morales Meseguer, Clínica Odontológica, Marqués del los Vélez s/n, 30008 Murcia, Spain; 2Faculty of Medicine and Odontology, Biomedical Research Institute (IMIB-Arrixaca) Hospital Morales Meseguer, Clínica Odontológica, Marqués del los Vélez s/n, 30008 Murcia, Spain; 3Departamento de Anatomía Humana y Psicobiología, Faculty of Medicine and Odontology, Biomedical Research Institute (IMIB-Arrixaca), University of Murcia Spain, 30100 Murcia, Spain

**Keywords:** oral leukoplakia, CO_2_ laser, recurrence, oral squamous cell carcinoma

## Abstract

**Simple Summary:**

The treatment of oral potentially malignant disorders (OPMDs), including oral leukoplakia (OL), is controversial. Medical interventions currently used to prevent malignant transformation in OL include surgical treatment, photodynamic therapy, and chemotherapy. The main advantages of laser surgery are the selective removal of the lesion, minimal damage to surrounding healthy tissue, and excellent postoperative wound healing. However, no treatment has been shown to prevent recurrence or significantly reduce malignant development in long-term follow-up studies, so further research is needed to identify possible risk factors.

**Abstract:**

Aim: The aim of this study is to assess the efficacy of CO_2_ laser treatment in oral leukoplakia and to analyse the recurrence rate of oral leukoplakia lesions at 18-month follow-up. Materials and methods: A prospective clinical study regarding CO_2_ laser treatment for oral leukoplakia was conducted, in which 39 patients with a total of 53 oral leukoplakias were included. Follow-up was performed at 18 months post-surgery and the following variables were studied: sex, age, associated risk factors, clinical classification, size, location and presence of epithelial dysplasia, recurrence, and rate of malignant transformation after resection. Results: In the analysis of the final results 18 months after baseline, a treatment success rate of 43.75% was observed. Oral leukoplakia recurred in 54.17% of cases, and 2.08% of leukoplakias progressed to cancer. Among all the studied variables (age, tobacco use, size, location, clinical type or histology), no significant differences were found with regard to recurrence. Conclusion: The use of CO_2_ laser therapy to treat leukoplakia lesions is sufficient to remove such lesions. However, parameters that can assess recurrence need to be sought.

## 1. Introduction

The management of oral potentially malignant disorders (OPMD), including oral leukoplakia (OL), is controversial [1,2,3]. Among OPMD, oral leukoplakia remains the most commonly encountered condition in clinical practice [4]. Recently, the working group on potentially malignant disorders coordinated by the WHO Collaborating Centre for Oral Cancer/Precancer presented a new definition of leukoplakia: “A white plaque of questionable risk having excluded (other) known disease or disorders that carry no increased risk for cancer” [5].

As stated in a systematic review by Petti et al. 2003 [3], the worldwide prevalence of leukoplakia ranges between 1.49–2.6%, suggesting that the global number of oral cancer cases associated with oral leukoplakia is likely underreported, particularly in South-East Asian countries and in India. The major aetiological factor in oral leukoplakia is tobacco use [1]. In a review by González-Moles et al. (2018) [6], it was observed that smokers exhibit a seven-fold greater risk of developing leukoplakia compared to non-smokers, and that this figure progressively increases in relation to the number of years of consumption and the amount of tobacco used on a daily basis. In this review, between 30% and 56% of lesions disappeared after three months of withdrawal. The role of alcohol in the development of oral leukoplakia is less clear [7,8,9,10,11]. Some authors agree that the combination of tobacco and alcohol does increase the risk of malignant transformation, and that this is probably related (1) to the increased solubility of carcinogens and atrophy of the oral mucosa caused by alcohol, (2) to the mutagenic agents generated in metabolism (acetaldehyde), and (3) to the decreased DNA capacity associated with alcohol consumption [2,10].

Oral leukoplakias occurring in the absence of such identifiable risk factors are described as idiopathic leukoplakias and are considered to have an underlying genetic basis [5]. The clinical features of oral leukoplakia may take different forms, defined in accordance with the clinical pattern (homogeneous or non-homogeneous), the distribution or spread of the lesion (focal or disseminated), and their location within the oral cavity [8,12].

The malignant transformation rate in oral leukoplakia varies from one site to another within the mouth, from one population to another, and from one group to another. According to the reviewed articles, it is estimated to range between 0.13% and 3.5% [13,14,15,16,17,18]. Clinical appearance is a key factor, since non-homogeneous leukoplakia exhibits a malignant transformation potential 4–5 times higher than that of homogeneous leukoplakia. In a study by Holmstrup et al. (2006) [15] involving 269 oral leukoplakia lesions, logistic regression analysis showed a seven-fold increased risk (Odds ratio = 7.0) of malignant development for non-homogeneous leukoplakia compared to homogeneous leukoplakia. The risk of malignisation in proliferative verrucous leukoplakia is very high: 65.8% [18,19,20,21,22].

Histopathological diagnostic criteria for leukoplakia range from simple epithelial hyperplasia with hyperparkeratosis or hyper(ortho)keratosis to epithelial dysplasia of variable severity. A recent meta-analysis by Iocca et al. (2020) [23] of published articles showed that moderate/severe dysplasia is significantly associated with a much greater risk of malignant transformation compared to mild dysplasia, with an odds ratio (OR) of 2.4. Mild dysplasia exhibited an annual malignant transformation of 1.7%, whereas severe dysplasia showed a rate of 3.57%.

The medical interventions currently used to prevent malignant transformation in OL include surgical treatment (cryotherapy, laser treatment, and cold-knife surgery), photodynamic therapy, and chemotherapy (vitamin A and retinoid, beta carotene or carotenoids, non-steroidal anti-inflammatory drugs and herbal extracts, bleomycin, and Bowman–Birk inhibitor) [24,25,26,27,28,29,30]. Nevertheless, no treatment has been shown to prevent recurrence or to significantly reduce malignant development in long-term follow-up studies [26,29]. Given that laser surgery is associated with low intraoperative and postoperative complication rates, laser surgery is one of the most common surgical treatments for OL. The major advantages of laser surgery are selective lesion removal and minimal damage to surrounding healthy tissues, excellent postoperative wound healing, and no visible scarring. In addition, laser surgery can be performed on an outpatient basis. The outcomes of studies investigating leukoplakias treated with CO_2_ laser are highly variable, especially in terms of recurrence and malignant transformation [30,31,32,33,34]. 

The objective of this study is to assess the efficacy of CO_2_ laser treatment in oral leukoplakia and to analyse the factors involved in the recurrence rate of oral leukoplakia lesions 18 months after vaporisation.

## 2. Material and Methods

This prospective clinical study was conducted at the Clínica Odontológica Universitaria, belonging to Hospital Morales Meseguer in Murcia (Spain). Thirty-nine patients with a total of 53 oral leukoplakias were included. All participants underwent the same procedures throughout the trial with no deviation from the protocol, were informed about the treatment, and gave written informed consent. The inclusion criterion was the presence of oral leukoplakia, as confirmed by histopathological study after exclusion of other diseases (leukoedema, frictional keratosis, and having eliminated risk factors). Patients initially diagnosed with oral squamous cell carcinoma (OSCC), erythroleukoplakia, or proliferative verrucous leukoplakia were excluded, in accordance with the ethical guidelines of the Declaration of Helsinki by the World Medical Association. The study required the approval of the Research Ethics Committee of the University of Murcia (ID: 2225/2018). A medical and dental history was taken, and possible aetiological factors were identified. All patients were treated with surgical resection and counselled in relation to tobacco use and alcohol consumption habits. The study collected data on habits (alcohol, tobacco), body mass index, oral hygiene (good/improvable) and bruxism (yes/no), lesion characteristics, clinical form, location, and type of mucosa and size. In the histopathological study, lesions were examined by the same pathologist. Histopathological diagnostic criteria for leukoplakia range from simple epithelial hyperplasia with hyperparkeratosis or hyperorthokeratosis to epithelial dysplasia of variable severity [5]. Epithelial dysplasia was scored according to the World Health Organization (WHO) classification scale. Regardless of whether dysplasia was present or not, a binary dysplasia scale (high or low grade) was used for the analysis.

In our study, both the clinician and patients followed the appropriate safety guidelines for laser use: protective goggles, limited access to the surgical area, and the utilisation of wooden tongue depressors for tissue separation and protection. The locations of lesions were as follows: 26 on the gingiva, 10 on the buccal mucosa, 7 on the hard palate, 4 on the tongue and 1 on the floor of the mouth.

In all patients, the surgical procedure was performed under local anaesthesia (4% articaine with 1:100,000 epinephrine as first choice and 3% mepivacaine if vasoconstrictors were contraindicated). Patients were asked to perform a preoperative rinse with 0.2% chlorhexidine as an antiseptic method.

In all cases, the surgical treatment to remove the lesions was CO_2_ laser (Lasersat 20 W, Satelec^®^, Barcelona, Spain, Pierre Rolland, SATELEC^®^, S.A., Barcelona, Spain) in continuous mode at a distance of approximately 10 mm from the oral mucosa and at intervals of approximately 5–10 s. Lesions were eliminated with the laser using vertical and horizontal movements (including 3 mm of clinically normal mucosa at the periphery).

Regarding the power of CO_2_ laser, our choice was based on published studies on laser-treated leukoplakia by Monteiro et al. 2017 [30], who used a power range of 5–10 W, depending on the operator’s criteria. After the surgery, gauze impregnated with 0.2% chlorhexidine was applied to the wound for 20 min, and tissue regeneration was through secondary intention. Postoperative instructions were provided to all patients, as well as chitosan-chlorhexidine 0.2% post-intervention topical gel (ISDIN Bexident^®^ Post Treatment Topical Gel, Barcelona, Spain), to be applied three times per day for 10 days. (Figure 1, Figure 2 and Figure 3). The treatment and subsequent check-up and follow-up appointments were conducted by the same operator. Post-surgery complications were recorded, and perceived pain was assessed by utilising the visual pain scale from 1–10 (with 1 meaning no pain and 10 the worst pain) one week after treatment. Follow-up periods were performed at 6 months, 12 months and 18 months post-surgery. Recurrence was defined as the reappearance of OL at the surgical site, whereas if no visible changes were observed in the oral mucosa of the treated area, the lesion was considered to have been successfully resolved.

**Statistical analysis:** The results were expressed as absolute and relative frequencies. The relationship between the oral leukoplakia location and clinical form was analysed by univariate analysis using the chi-square test. The differences were considered as statistically significant at *p*-value < 0.05. In order to determine the effect of lesion-related variables (clinical form, location, size, and dysplasia) and the variables regarding patient habits (tobacco use, oral hygiene, body mass index, and bruxism) on recurrence, univariate logistic regression models were applied. Cumulative recurrence of the lesion at 18 months was analysed with the Kaplan-Meier method by creating a disease-free survival curve.

## 3. Results

Thirty-nine (39) patients with a total of 53 oral leukoplakia cases were included in the study (5 patients, however, were excluded from the study because they did not give consent/could not attend the follow-up appointments due to transportation difficulties). Thus, the final study sample consisted of a total of 34 patients with 48 oral leukoplakias treated by means of CO_2_ laser. Female patients represented 52.1% (*n* = 25) of leukoplakias and male patients 47.9% (*n* = 23), with ages ranging from 48–81 years and an average age of 61.3 years (SD = 10). In 35.4% of cases, oral hygiene was considered as “improvable” because of the presence of bacterial plaque, while in 64.6% of cases, oral hygiene was considered as “good”; this was assessed by the same operator. None of the patients reported daily alcohol consumption. Seventy-five percent of the lesions occurred in non-bruxing patients. In terms of oral prostheses, 10.41% were removable partial denture wearers, 25% had fixed dentures, and 31.25% had at least one implant-supported crown.

The clinical form non-homogeneous leukoplakia is the most common, since it represents a higher percentage (62.5%) in comparison with homogeneous leukoplakia (37.5%).

In 54.2% of cases, the lesion was less than 2 cm in size. In our histopathological assessment, 37 (77.1%) of the studied oral leukoplakias showed no dysplasia. Dysplasia was present in 11 lesions (10 with low grade and 1 with high grade). The location of the lesion was not associated, in a statistically significant manner, with the homogeneous/non-homogeneous clinical form (*p*-value = 0.107).

One week after CO_2_ laser vaporisation, follow-up was performed, and the grade of pain was assessed on a scale from 1–10. Symptomatology was minimal in most cases; only 10.4% presented a value higher than 6. Post-surgical complications included mild bleeding in two cases and superinfected lesion in two cases. In the analysis of the final results, 18 months after the beginning of treatment, a 43.75% success rate was observed regarding the treatment of oral leukoplakia with CO_2_ laser, i.e., no pathology was observed in the treated area during final follow-up. Oral leukoplakia did recur in 54.17% of cases, and 2.08% of the treated leukoplakia cases progressed to oral squamous cell carcinoma. To determine the effect of lesion-related variables and the variables regarding patient habits on recurrence, univariate logistic regression models were applied; the results are shown below in Table 1 and Table 2. The results demonstrated that none of the lesion-related variables had a statistically significant effect on the course of recurrence (Table 1). In relation to the variables regarding patient habits (Table 2), the results also showed no statistically significant effect on the course of recurrence. As seen in the survival curve developed by means of Kaplan-Meier method, patient disease-free survival at 18 months was 23.3 ± 0.06, with an estimated 10.2 ± 0.8 months (95% CI = 8.63–11.8) on average until recurrence of the lesion (Figure 4).

## 4. Discussion

The treatment of oral leukoplakia is a real challenge due to recurrence rate and malignant transformation. This prospective clinical study showed that at 18 month follow-up of oral leukoplakia cases treated with CO_2_ laser, the recurrence rate was 54.17% and the malignant transformation rate was 2%.

Considering the question “does laser resection of oral leukoplakia impact recurrence and malignant transformation?”, a systematic review and meta-analysis by Pauli Paglioni et al. (2020) [32] showed that surgical laser ablation of OL can reduce recurrence rates, and that this therapy has no effect on malignant transformation when compared to conventional treatments. Laser—unlike cold-knife surgery—presents a series of advantages, such as instant sterilisation of the surgical wound, better visualization, non-contact surgery and, therefore, no mechanical trauma to tissues [24]. Other advantages are healing through secondary intention, reducing the duration of surgery and the distribution and depth of scars, as well as less pain and inflammation in the treated area. Thus, in this study, in only 10.4% of CO_2_ laser vaporisations, patients felt more than grade 6 pain in the first days post-surgery.

Our results show that in patients with OL, the variables of tobacco, clinical form, location, and size do not indicate a higher risk of recurrence after surgery [17,30].

Determining the margins in OL during surgery remains a challenge for lesion recurrence. In a study by Kuribayashi et al. [35], a significant correlation between surgical margins and OL recurrence after surgery was reported. A lower recurrence rate in patients with >3 mm-wide resection margins was found; those authors suggested these margins to be the optimal safety margins. A clinical trial by Romeo et al. [28] observed a 45.5% recurrence rate in the treated group without margins versus a 36.4% recurrence rate in the treated group with at least a 3 mm margin after 6 months of treatment.

Similarly, a study by Vilar-Villanueva et al. [30] included 58 patients with a mean follow-up time 57.5 months and recurrence rate of 52.6%. Among all the studied variables, margin was the only one for which a statistically significant correlation with lesion recurrence was demonstrated.

Optical adjunctive aids may be one way to to determine the margins of OL. Tiwari et al. [36] conducted a review of the efficacy of direct optical fluorescence imaging as an adjunct to comprehensive oral examination in the clinical evaluation and surgical management of OL, and concluded that optical fluorescence can provide indications for determining surgical margins.

Mogedas-Vegara et al. [27] found no statistically significant correlation between multiple lesions and recurrence or malignant transformation. In contrast, for proliferative verrucous leukoplakia, other authors [20,21] described recurrence rates of 67.2% and 65.8% respectively, and determined there is insufficient scientific evidence to conclude that no treatment strategy is capable of reducing recurrence. In this study, OL cases which transformed into OSCC experienced recurrence prior to OSCC transformation, resulting in a significantly higher risk of malignant transformation for patients with recurrent OL in comparison with those with non-recurrent OL.

In a systematic review and meta-analysis, Dong et al. [16] studied the malignant transformation rate of oral leukoplakia treated with carbon dioxide laser. A total of 1546 patients with 1864 lesions were included, and the overall rate of malignant transformation was found to be 4.50% (95% CI 0.0305–0.0659). All leukoplakia cases, and not only those with dysplasia, should be considered to be at risk of progressing to OSCC. It is clearly recognised that patients diagnosed with OL who possess the following characteristics are at increased risk of cancerisation: advanced age, female sex, leukoplakia greater than 200 mm^2^, non-homogeneous type (e.g., erythroleukoplakia), the highest grades of dysplasia, etc. [16,37,38] However, to date, there is no generally approved standard systemic therapy regimen to treat oral leukoplakia in order to prevent oral cancer.

It is important to keep in mind that most studies monitoring recurrence rates of OL after surgical excision are retrospective in design, and the results are difficult to compare due to differences in designs, inclusion and exclusion criteria, treatment interventions, surgical techniques, or follow-up times.

According to Holmstrup and Dabelsteen [38], an important aspect is the observation of cancer development, even after surgical removal of clinical lesions. Intensive leukoplakia follow-up programs are important, regardless of surgical intervention, as cancer transformation occurs in 3–11% of cases at the site of the resected lesion despite surgical treatment [38,39,40]. There is another point to consider, i.e., surgery as an invasive treatment. This may increase the risk of malignant transformation based on the ‘field cancerisation concept’. It is easy to excise tissue whose appearance has changed, but it is very difficult to eradicate all genetically altered cells, since there is genomic instability throughout the epithelium. During the postoperative wound healing process, proliferation capacity of residual altered cells is better than that of normal mucosal cells [40], and the molecular signature generated in the microenvironment may stimulate cancer growth.

In addition, limitations should be taken into consideration. Firstly, as in the majority of such studies, in our research, there was no control group because it is considered unethical to leave any patient untreated due to the aforementioned high probability of progression to cancer compared to healthy patients. We must also consider the small sample size as a limitation of the study. Additionally, the follow-up period was 18 months, and longer follow-ups are required to assess the malignant transformation rate.

## 5. Conclusions

Despite the fact that an effective treatment for OL has not yet been developed, CO_2_ laser vaporisation is a valid treatment, with a low rate of postoperative complications, although it is not a guarantee against recurrence. Thus, further long-term prospective clinical studies are essential. The assessment and study of molecular biomarkers could be a potential monitoring tool to filter patients at risk of developing oral cancer and to follow them up more closely.

## Figures and Tables

**Figure 1 cancers-14-05455-f001:**
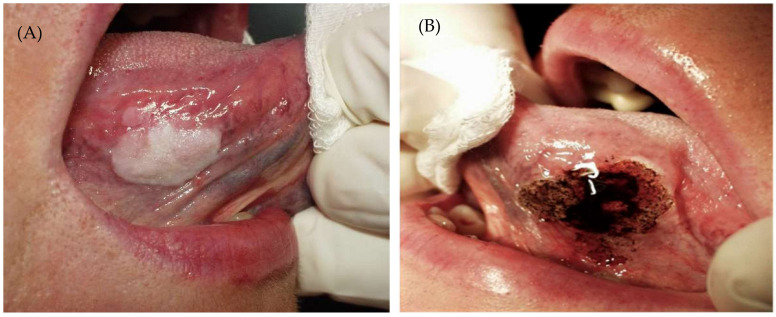
(**A**). Oral leukoplakia Before CO_2_. (**B**) Immediately after treatment.

**Figure 2 cancers-14-05455-f002:**
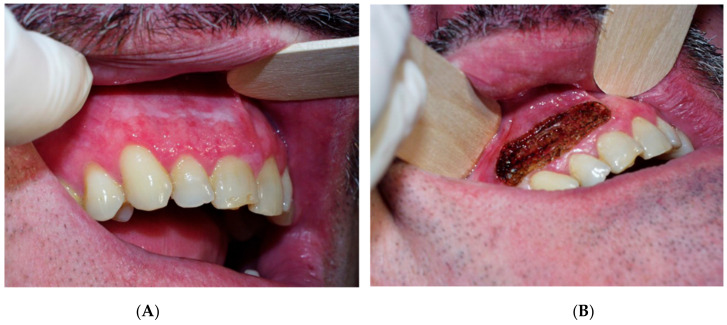
(**A**) Lesion before CO_2_. (**B**) Immediately after carbon dioxide surgery.

**Figure 3 cancers-14-05455-f003:**
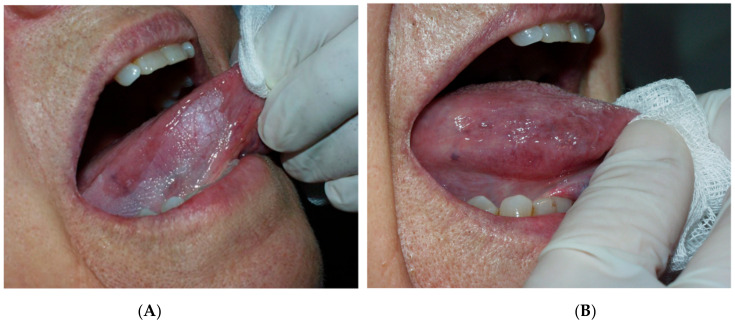
(**A**) before removal with CO_2_ laser on lingual margin and (**B**) 3 months after treatment.

**Figure 4 cancers-14-05455-f004:**
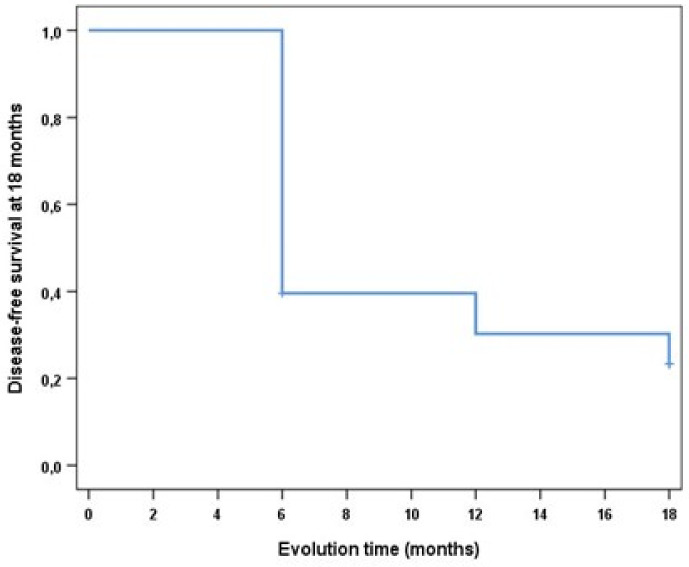
Survival curve developed by the Kaplan-Meier method. Patient disease-free survival at 18 months was 23.3 ± 0.06, with an estimated 10.2 ± 0.8 months (95% CI = 8.63–11.8) on average until recurrence of the lesion.

**Table 1 cancers-14-05455-t001:** CO_2_ laser treatment of oral leukoplakia:recurrence.

	Recurrence, *n* (%)		Univariate Logistic Regression		
	No	Yes	OR	IC 95%	*p*-Value
Clinical type					
Leucoplakia Homogeneous	6 (33.3)	12 (66.7)	1		
Leucoplakia non-homogeneous	15 (50)	15 (50)	2	0.59–6.73	0.263
Location					
Lining Mucosa	6 (46.2)	7 (53.8)	1		
Masticatory Mucosa	13 (39.4)	20 (60,6)	1.319	0.36–4.81	0.675
Size					
<2 cm	11 (42.3)	15 (57.7)	1		
>2 cm	10 (45.5)	12 (54.5)	0.88	0.28–2.76	0.827
Dysplasia					
No	14 (37.8)	23 (62.2)	1		
Yes	7 (63.6)	4 (36.4)	0.348	0.09–1.41	0.138

OR: odds ratio. IC: confidence Interval.

**Table 2 cancers-14-05455-t002:** Effect of variables related to patient habits on recurrence.

	Recurrence, *n* (%)		Univariate Logistic Regression		
	No	Yes	OR	IC 95%	*p*-Value
Tobacco					
No	10 (34.5)	19 (65.5)			
Yes	9 (81.8)	2 (18.2)	0.163	0.02–1.40	0.098
former smoker	2 (25)	6 (75)	1.579	0.27–9.31	0.614
Oral hygiene					
Good	14 (45.2)	17 (54.8)			
Bad	7 (41.2)	10 (58.8)	1.176	0.36–3.90	0.790
Diabetes					
No	19 (50)	19 (50)			
Yes	2 (20)	8 (80)	4	0.75–21.35	0.105
Body mass index					
Normal weight	4 (28.6)	10 (71.4)			
overweight	17 (50)	17 (50)	0.400	0.11–1.53	0.180
Bruxism					
No	17 (47.2)	19 (52.8)			
Yes	4 (33.3)	8 (66.7)	1.789	0.46–7.02	0.404

OR: odds ratio. IC: confidence interval.

## Data Availability

The authors hereby confirm that all the data of this research are available within this manuscript.

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
