# Peer review of "Recurrence of Oral Leukoplakia after CO2 Laser Resection: A Prospective Longitudinal Study"

_cancers, 2022, doi:10.3390/cancers14215455_

Round 1

Reviewer 1 Report

1, In the first place, what is the benefit Laser surgery for oral leukoplakia. Please describe what one thinks is good compared with other surgery or only observation in Introduction or Discussion.

2, The authors should describe pathological diagnosis, eg: hyper keratosis, parakeratosis, acanthosis. It is very important that is not only dysplasia but also other pathological diagnosis. Because Leukoplakia is not pathological diagnosis.

3, In page 13 and 12, the same table is described. Is it wrong?

4, The authors should study with relevant other confounding factor. Did you have multivariate analysis data?

Author Response

We would like to thank both our reviewers and the editors for their encouraging and considered comments. We believe that responding to each of the issues raised has made the paper considerably clearer and stronger. Significant changes have been marked in yelow in the revised manuscript.

I carefully read the manuscript and commented my considerations for the study. I hope it would be a help in improving your manuscript

Referee 1

1, In the first place, what is the benefit Laser surgery for oral leukoplakia. Please describe what one thinks is good compared with other surgery or only observation in Introduction or Discussion.

Thank you for your suggestion. We have added the detailed  introduction and discussion

clinical Laser—unlike cold-knife surgery—presents a series of advantages, such as instant sterilisation of the surgical wound, enables better visualization, entails non-contact surgery and, therefore, no mechanical trauma to tissues[24]. Another advantage to be considered is healing through secondary intention, reducing the duration of surgery and the distribution and depth of scars, as well as less pain and inflammation in the treated area

2, The authors should describe pathological diagnosis, eg: hyper keratosis, parakeratosis, acanthosis. It is very important that is not only dysplasia but also other pathological diagnosis. Because Leukoplakia is not pathological diagnosis.

We agree 

We followed criteria 

  • Malignant transformation of OL is associated with a variety of clinical and histological features, including: a previous history of cancer diagnosed in the head and neck region; advanced age; clinical appearance, size, anatomic site of the lesion; and, most importantly, the degree of epithelial dysplasia. Rubert A, Bagán L, Bagán JV. Oral leukoplakia, a clinical-histopathological study in 412 patients. J Clin Exp Dent. 2020 Jun 1;12(6):e540-e546.
  • Evren I, Brouns ER, Poell JB, Wils LJ, Brakenhoff RH, Bloemena E, de Visscher JGAM. Associations between clinical and histopathological characteristics in oral leukoplakia. Oral Dis. 2021 Oct 3
  • Warnakulasuriya S, Kujan O, Aguirre-Urizar JM, Bagan JV,González-Moles MÁ, Kerr AR, et al. Oral potentially malignant disorders: a consensus report from an international seminar on nomenclature and classifcation, convened by the WHO Collaborating Centre for Oral Cancer. Oral Dis 2021; 27(8),1862–1880.https://doi.org/10.1111/odi.13704

3, In page 13 and 12, the same table is described. Is it wrong?

 Has been changed

4, The authors should study with relevant other confounding factor. Did you have multivariate analysis data?

Thank you and the observation is very interesting. We did not do it because none of the variables studied was significant.

Thank you and the observation is very interesting. We did not do it because none of the variables studied was significant.

Reviewer 2 Report

This research studied laser ablation of leucoplakia. Leucoplakia is a common disease, so it is good that new treatment strategies are being proposed. The main point of the research is interesting, but the description of material and method is insufficient.

Material and methods section

The description of the site of leukoplakia should be more detailed. For example, tongue, gingiva, buccal mucosa, etc.

I would like to confirm a more detailed description of the excision methods. As mentioned in the discussion, please indicate whether you have decided on a safety margin for the resection. Also, did you use a biological staining method such as iodine staining? If you used them, please describe them.

Results section

Tables 1 and 2 are identical.

Homogeneous leukoplakia showed a higher recurrence rate. Please explain your opinion about this result.

Discussion

The number of participants is relatively few. Please describe the limitation of the study in the discussion.

Author Response

this research studied laser ablation of leucoplakia. Leucoplakia is a common disease, so it is good that new treatment strategies are being proposed. The main point of the research is interesting, but the description of material and method is insufficient.

Thank you for your comments I carefully read the manuscript and commented my considerations for the study. I hope it would be a help in improving your manuscript

Material and methods section

The description of the site of leukoplakia should be more detailed. For example, tongue, gingiva, buccal mucosa, etc. It has been completed  

I would like to confirm a more detailed description of the excision methods. As mentioned in the discussion, please indicate whether you have decided on a safety margin for the resection. Also, did you use a biological staining method such as iodine staining? If you used them, please describe them.

Thank you for your suggestion.

 We have added the detailed clinical characteristics of patients enrolled in this study

Eearly detection of dysplasia in potentially malignant oral disorders (PMD) could facilitate screening for possible subsequent malignant transformation. Vital staining (methylene blue (MB) and Lugol's iodine (LI)) is a non-invasive clinical adjunct used to determine the biopsy site, which facilitates early detection of dysplastic changes in the OL and also aids in the detection of margins.In the present study, we have not used stains

Results section  Tables 1 and 2 are identical. It has been changed

Homogeneous leukoplakia showed a higher recurrence rate. Please explain your opinion about this result.

There is  explanation, perhaps due to the sample size; future studies with a larger population are needed

Discussion . We agree .The number of participants is relatively few. Please describe the limitation of the study in the discussion.

We have mentioned this in the Limitation

Reviewer 3 Report

 The prospective clinical study is well carried on , but:

- In Materials and Methods you write the  laser parameters, used by Monteiro between 5 and 10 watts in cw at a distance of 10mm, but you should specify which power set you applied in your clinical study ( lines 121-126)

- You could show one more clinical figure before ,immediately after laser treatment, at control T1( after 1 week), T2 (after 1 month), T3 (after 18 months)

Author Response

The prospective clinical study is well carried on , but:

Thank you for your comments

- In Materials and Methods you write the  laser parameters,  It has been added

- You could show one more clinical figure before ,immediately after laser treatment, at control T1( after 1 week), T2 (after 1 month), T3 (after 18 months)

 Clinical Images have been added in different periods

Round 2

Reviewer 1 Report

Thank you for the authors comments and revised manuscript.

 Are pathological diagnoses only dysplasia in this study? Please more describe criteria for diagnosis of leukoplakia. 

Author Response

Thank you for your comments have been added

The histopathologic diagnostic criteria for leukoplakia range from simple epithelial hyperplasia with hyperparkeratosis or hyper(ortho)keratosis, variable severity of epithelial dysplasia. 

The most recent definition by the WHO Collaborating Centre, published in 2007, was “A predominantly white plaque of questionable risk having excluded (other) known diseases or disorders that carry no increased risk for cancer” (Warnakulasuriya et al., 2007).

Reviewer 2 Report

The research improved significantly. Laser treatment for oral leukoplakia has been reported infrequently and is therefore useful.

Author Response

Thank you for your comments